# Prospective Analysis of the Temporal Relationship between Psychological Distress and Atopic Dermatitis in Female Adults: A Preliminary Study

**DOI:** 10.3390/healthcare10101913

**Published:** 2022-09-29

**Authors:** Gurkiran Birdi, Michael Larkin, Rebecca C. Knibb

**Affiliations:** School of Psychology, College of Health and Life Sciences, Aston University, Birmingham B4 7ET, UK

**Keywords:** atopic dermatitis, stress, longitudinal, psychodermatology

## Abstract

Atopic dermatitis (AD) has been associated with psychological distress, but few studies have examined the causal relationships. This study aimed to investigate whether stress, anxiety, or depression could lead to an increase in AD severity or vice versa in adults using a longitudinal study design. Daily diaries measuring psychological stress were completed over four weeks; validated questionnaires measuring stress, anxiety, depression, and AD severity were completed weekly for twelve weeks. Thirty-six participants (all female, aged 18–46 years) were recruited; complete data were returned from 19. Stress and disease severity were significantly correlated when measured daily and weekly for the duration of the study. Cross-lagged panel model (CLPM) analyses identified that for the weekly measures, stress, anxiety, and depression on week X significantly predicted disease severity on week X + 1. Disease severity on week X also predicted psychological stress, anxiety and depression on week X + 1 for the majority of the twelve weeks. There appears to be a bi-directional relationship between stress, anxiety and depression, and AD severity in women. High levels of distress should be identified so that optimum management strategies can be implemented to reduce the risk of increased AD severity and the resulting impact severity might have on psychological wellbeing.

## 1. Introduction

Atopic dermatitis (AD) is a chronic inflammatory skin disease characterized by intense pruritus, erythematous lesions, increased water loss through the skin, and dry skin. The lifetime prevalence of AD is 10–20% in children and 1–3% in adults worldwide, with 85% of affected children developing the disease before the age of 5 years [1,2]. Although AD can resolve as children get older, it can persist into or manifest for the first time in adulthood [3]. Less than half of AD patients show complete resolution by the age of seven and only 60% by adulthood [4].

AD severity has been shown to be significantly related to psychological stress. In cross-sectional retrospective studies, psychological stress has been reported to exacerbate AD, with patients experiencing a worsening of symptoms following exposure to stress [5,6,7,8,9], possibly due to the activation of the hypothalamic–pituitary–adrenal (HPA) axis affecting the epidermal barrier and lowering the itch threshold [10,11,12]. However, the retrospective nature of the studies means that the results may be affected by recall bias, with recall in these studies ranging from a week to a year. Major stressful events may be better recalled, such as an earthquake [13,14]; however, minor daily stressful events that could contribute to the exacerbation of AD may be forgotten with time when conducting a retrospective study. In addition, most of the studies examining the association between stress and AD have focused on children and adolescents and there are few that explore this is adults [15,16].

Only one known study, published over 20 years ago [17], has examined the relationship between AD severity and stress longitudinally in adults, using daily diaries over two-weeks with 50 participants. They found that interpersonal stress on day X significantly predicted the severity of AD on day X + 1 using lag-sequential analyses. This relationship was reciprocal, as AD severity on day X also indicated stress level on day X + 1. The contribution made by psychosocial stress in maintaining and exacerbating AD is also supported by the effectiveness of psychotherapeutic interventions that incorporate relaxation techniques or stress management, which have led to the significant improvement of AD symptoms [18].

Due to the lack of recent longitudinal studies in this area and the fact that there are no studies examining other areas of mental health, the aims of this study were to investigate whether stress, anxiety, or depression could lead to an increase in AD severity or vice versa in adults using a longitudinal study design. The previous longitudinal study on adults with AD suggested that stress and AD severity can have an impact on each other in a short period of time, and so daily diaries were used to document AD severity and stress, as recommended [19,20]. Retrospective studies have asked people to recall stressful events over much longer periods of time, and so it was decided to run a longitudinal study across 12 weeks, with weekly measures taken in addition to the daily diaries. No similar studies on adults with AD have been published, and the ability to recruit participants to a study using this design, and completion and drop-out rates were monitored.

## 2. Materials and Methods

### 2.1. Study Design

This study used a quantitative prospective design with validated measures of stress, anxiety, depression and AD severity, completed weekly for 12 weeks alongside daily stress and symptom severity diaries for four weeks. This study received approval from the University Research Ethics Committee (application no. 1428) and all participants provided informed consent before taking part.

### 2.2. Participants

Participants had to have a clinical diagnosis of AD, not be undergoing any medical or therapeutic treatment for a mental health condition, and to be aged 18 years or over to be eligible for the study. Participants were recruited from AD support groups via advertising on their social media sites. Advertisements were posted for a period of one month and the ability to recruit to this study design within this time period was monitored.

### 2.3. Measures

Socio-demographic information was collected such as age, gender, occupation and ethnic background. AD-specific information included diagnosis, medication and family history.

Subjective disease severity was measured using the Patient Oriented Eczema Measure (POEM) [21], a subjective disease severity measure that is widely used in AD research. It focuses on the illness as experienced by the patient and has five severity bandings, with a higher score related to more severe disease. The POEM was completed by participants at the end of each week for 12 weeks and had high internal consistency for this sample (alpha = 0.82).

Psychological stress was measured using the Perceived Stress Scale (*PSS-14*), a 14-item validated questionnaire [22] that measures the degree to which one’s situations in life are appraised as stressful. A higher score on this scale relates to higher stress levels. The PSS was completed at the end of each week for 12 weeks and had high internal consistency for this study (alpha = 0.85).

Anxiety and depression levels were assessed using the Hospital Anxiety and Depression Scale (HADS) [23]. The HADS consists of fourteen items, seven of which relate to anxiety and seven focus on depression. A higher score relates to greater anxiety or depression. The HADS internal consistency for this study sample was high for both anxiety and depression subscales (0.82 and 0.75, respectively). This scale was completed once weekly for the 12-week period.

A modified version of King & Wilson’s diary was used [17]. The diary consisted of a series of questions that assessed interpersonal stress and responses were rated on a 3-point scale (0—no, 1—a little, 2—a lot). Participants were also asked to state what AD related symptoms they had experienced and rate AD severity on a three-point scale (0—mild, 1—moderate, 2—severe). Participants were asked to complete these diaries each night for four weeks prior to retiring for sleep.

### 2.4. Procedure

Participant information sheets, consent forms, diaries and questionnaires were posted to home addresses of participants who responded to the study advert. Pre-paid envelopes were provided for participants to send completed packs back to the researcher. Participants completed daily and weekly measures for 12 weeks. All participants were added to a study-specific Facebook group and were sent weekly reminders to complete the measures.

### 2.5. Data Analyses

Statistical analyses were conducted using SPSS for Windows (version 23, SPSS, Chicago, IL, USA). Only complete datasets were included in the analysis. Correlational analyses were conducted between AD severity and psychological variables for each week. Additionally, a cross-lagged panel model (CLPM) analysis was conducted to examine the ability of interpersonal stress to predict skin symptoms on the next day for each participant and vice versa.

## 3. Results

### 3.1. Completion and Drop-Out Rates

Thirty-six participants responded to the study advert over the one-month recruitment period and were sent study packs, out of which 21 participants returned packs to the researcher. Nineteen participants out of 21 had complete data (a total of 4104 data points), with no missing data for the weekly or daily questionnaires (52% completion rate). Further drop-out analysis was not possible as the 15 participants who did not complete the study did not return any information.

### 3.2. Participant Sample Characteristics

All participants reported being clinically diagnosed with the condition. The participant sample consisted of all females, 14 (73.7%) were White and five (26.3%) were Asian. Eight participants were aged between 25–39 (42.3%), eight participants were aged 40–46 (42.3%), 3 were aged between 18–24 (15.4%). Fifteen participants (*n* = 15, 78.9%) reported being diagnosed with AD by a dermatologist and four participants (*n* = 4, 21.1%) were diagnosed by their GP. Fifteen participants (*n* = 15, 78.9%) reported family history of the condition. All 19 participants reported one or more concomitant atopic condition alongside their AD; 17 (89.5%) had hay fever, 14 (73.7%) reported asthma, and 11 (57.9%) reported suffering from food allergies. Eight (42.1%) reported suffering from all three conditions in addition to AD.

Weekly correlations and predictions between atopic dermatitis severity and psychological variables.

Pearson’s correlations were conducted for AD severity (POEM scores) and anxiety, depression and stress in the same week (Table 1). There were significant positive medium to large correlations between AD severity and anxiety (r’s = 0.47–0.75) for each week for the 12 weeks. Similarly, AD severity was significantly associated with depression for all 12 weeks apart from week 2; these correlations were also medium to large (r’s = 0.31–0.69). Psychological stress was also significantly correlated with AD severity for all 12 weeks, with correlations ranging from medium to large (rs = 0.55–0.72) (Table 1).

Three cross-lagged panel model analyses were conducted to explore the temporal relationship between the psychological variables and AD severity. When analysing cross-lagged model fit indices [24], a non-significant chi-square test is desired, which signifies that that the model approximates the underlying data and therefore should not be rejected [25]. Additionally, the Comparative Fit Index (CFI) is examined with possible values ranging from 0.00 to 1.00, with higher values signifying a better fitting model [26,27]. Further, the Root Mean Square Error of Approximation (RMSEA) is considered as this accounts for the error of approximation in the population; values less than 0.05 are indicative of good fit and values as high as 0.08 represent several errors of approximation in the population [28]. Lastly, the Tucker–Lewis Index (TLI) produces values that range from 0 to 1 [29], with higher values indicating a good fit [30].

The cross-lagged models indicated an average fit for stress and AD severity (χ^2^(1800) = 1677, *p* < 0.001; CFI = 0.54, TLI = 0.67, RMSEA = 0.43, 95% CI [0.42, 0.46]), anxiety and AD severity (χ^2^(1654) = 1066.98, *p* < 0.001; CFI = 0.32, TLI = 0.21, RMSEA = 0.12, 95% CI [0.09, 0.34]), and depression and AD severity (χ^2^(1435) = 1165.67, *p* < 0.001; CFI = 0.06, TLI = 0.02, RMSEA = 0.61, 95% CI [0.54, 0.67]).

When exploring the temporal relationship between psychological variables and AD severity (Table 2), AD severity on week X significantly predicted stress on week X + 1 in eleven out of twelve weeks. Stress on week X also significantly predicted AD severity on week X + 1 in eleven of the weeks. Similarly, bi-directional predictions were found for most of the 12 weeks when exploring AD severity, anxiety and depression. AD severity in week X predicted anxiety on week X + 1 in eleven out of twelve weeks, and anxiety on week X also predicted AD severity on week X + 1 in ten of the weeks. Moreover, AD severity in week X predicted depression on week X + 1 in ten out of the total twelve weeks and depression in week X predicted AD severity on week X + 1 in nine of the weeks.

Daily correlations and predictions between atopic dermatitis severity and psychological stress

Daily diary ratings of stress and AD severity were used to further examine causal relationships. The cross-lagged model (see Appendix A) results indicated that the model was an average fit (χ^2^(1769) = 10,557.87, *p* < 0.001; CFI = 0.052, TLI = 0.001, RMSEA = 0.52, 95% CI [0.51, 0.53]). AD severity on a given day significantly predicted stress the following day in only two of the 30 days (day 1: β= −0.54, *p* < 0.05, day 22 = β = 0.83, *p* < 0.01). On day one, higher AD severity predicted lower stress levels on the following day, whereas for day 22 higher AD severity predicted greater stress the following day.

Stress on day X predicted AD severity on day X + 1 for three out of the 30 days (day 5: β = −0.62, *p* < 0.001, day 22: β = 0.74, *p* < 0.001, day 29: β = −0.41, *p* < 0.05). On days five and 29, higher stress predicted less AD severity the next day whereas on day 22, higher stress predicted greater AD severity the next day. The covariances between stress and AD severity showed significant positive relationships for 23 out of 30 days; higher stress was correlated with increased AD severity (Table 3) and these correlations were medium to large-sized.

## 4. Discussion

This study aimed to investigate whether stress, anxiety, or depression could lead to an increase in AD severity or vice versa in adults using a longitudinal study design. The study has demonstrated a possible bi-directional relationship between psychological distress and AD severity in women which warrants further investigation in a larger study. This study found positive medium to large sized relationships between AD severity and psychological variables measured weekly, which supports findings from several cross-sectional studies [7,8,9]. Although there were no significant temporal relationships between psychological distress and AD severity on a daily basis, the lag sequential weekly analyses between psychological variables and disease severity revealed that both were predictive of the other, suggesting a bi-directional relationship.

Psychological stress has been widely researched in relation to AD, with several studies finding that patients with AD suffer from stress-related exacerbations [12,31,32]. This contributes to an itch–scratch cycle that results in a state of high anxiety and stress [31]. Much of present retrospective cross-sectional research also points to the existence of a bi-directional relationship [5,6,7,8,9], which is supported by the findings from the present longitudinal study. The difference in daily compared to weekly findings may suggest that the physiological effects of stress, anxiety, and depression need some lead time before resulting in disease exacerbation and vice versa, and so daily measurements may not reflect these changes. However, the small sample size might have contributed to the lack of significant findings in the daily measures. Future studies should therefore include both daily and weekly measures to explore these hypotheses.

In the present study, 36 participants were recruited; however, only 19 participants returned completed daily diaries and weekly questionnaires: a completion rate of 52%. This should be taken into account when planning future studies with this design. Recruitment and retention are challenging factors for researchers conducting longitudinal studies. For example, the more demanding or burdensome the study is, the worse the recruitment and retention rates are [33,34]. This study included daily diaries which may have added some burden although feedback provided spontaneously from six participants indicated that the diaries were not a cumbersome task. This is supported by the number of participants returning fully completed study packs with no missing data.

This study recruited from support groups via social media. There are limitations to this method of recruitment. Firstly, the degree of non-response bias could not be ascertained, as the researchers were unable to determine a response rate. Secondly, there is potentially a selection bias in favor of young adults with AD, as they are more likely to be Facebook users. Duggan and Brenner [35] found statistically significant differences among different age groups who were Facebook users, with 86% of all people aged 18–29 years old using the site compared to 73% of people aged 30–49, 57% of people 50–64, and only 35% of people older than 64 years old. Indeed, participants in this study were aged 18–49, with no participants over the age of 50. In addition to the above limitations, this study was not able to recruit any males, although evidence suggests the presence of sex differences in the skin–psyche relationship [8,36].

Due to the design of this study, AD severity was measured by self-report rather than through objective measures. Self-report of skin disease severity is common, particularly in cross-sectional community-based studies [37,38,39], and there is evidence for the validity of self-reported presence of skin disease and of self-reported skin symptoms as measures of objective disease presence [40]. It is also arguable that the self-perception of AD severity as measured by self-report is as important in order for it to be understood as a more objective measure of severity, when exploring its relationship with psychological distress. Nevertheless, objective measures of stress (such as taking salivary cortisol samples) and clinical ratings of disease severity could be used in future studies in order to further explore the physiological mechanisms underlying any causal relationships.

Despite these limitations, the current study provides preliminary evidence of a causal bi-directional relationship between psychological distress and AD severity which warrants further investigation. It would be useful within clinical settings to identify high levels of stress in patients with AD so that strategies can be implemented to reduce the risk of increased AD severity and so that severity does not have an impact on psychological well-being. The findings of this study also provide information for further research of this nature. We suggest that drop-out rates of approximately 50% should be taken into consideration when recruiting and that both daily and weekly measures of psychological distress and AD severity should be used. Daily diaries and weekly psychometric measures are feasible across a three-month period, but a range of recruitment sources should be utilized to reach a demographically diverse population.

## Figures and Tables

**Table 1 healthcare-10-01913-t001:** Pearson’s correlations between atopic dermatitis severity and psychological variables.

POEM	Anxiety	Depression	Stress
Baseline	0.55 *	0.47 *	0.63 *
Week 1	0.65 **	0.50 *	0.62 **
Week 2	0.47 *	0.31	0.52 *
Week 3	0.51 *	0.60 **	0.70 ***
Week 4	0.54 *	0.69 ***	0.55 *
Week 5	0.48 *	0.53 *	0.55 *
Week 6	0.55 *	0.52 *	0.69 **
Week 7	0.72 ***	0.59 **	0.72 ***
Week 8	0.69 ***	0.57 *	0.67 **
Week 9	0.59 **	0.69 ***	0.58 **
Week 10	0.55 *	0.51 *	0.60 **
Week 11	0.66 **	0.59 **	0.66 **
Week 12	0.75 ***	0.69 **	0.67 **

* *p* < 0.05, ** *p* < 0.01 *** *p* < 0.001.

**Table 2 healthcare-10-01913-t002:** Standardized parameter estimates for random-intercept cross-lagged panel models on psychological variables (stress, anxiety and depression) and AD severity (SEV) (*n* = 19).

Stress					
Cross-Lagged	β	SE	Cross-Lagged	β	SE
SEV0→STR1	1.41 ***	0.26	STR0→SEV1	0.45 ***	0.09
SEV1→STR2	135 ***	0.25	STR1→SEV2	−0.06	0.10
SEV2→STR3	1.51 ***	0.30	STR2→SEV3	0.76 ***	0.09
SEV3→STR4	0.65 **	0.20	STR3→SEV4	0.35 ***	0.09
SEV4→STR5	−0.18	0.18	STR4→SEV5	0.41 ***	0.11
SEV5→STR6	1.02 ***	0.22	STR5→SEV6	0.37 *	0.15
SEV6→STR7	0.89 **	0.28	STR6→SEV7	0.44 ***	0.10
SEV7→STR8	1.59 ***	0.15	STR7→SEV8	0.49 ***	0.07
SEV8→STR9	1.30 ***	0.26	STR8→SEV9	0.54 ***	0.08
SEV9→STR10	1.38 ***	0.18	STR9→SEV10	0.50 ***	0.10
SEV10→STR11	0.89 ***	0.22	STR10→SEV11	0.43 ***	0.08
SEV11→STR12	1.33 ***	0.15	STR11→SEV12	0.48 ***	0.08
**Anxiety**					
Cross-lagged	β	SE	Cross-lagged	β	SE
SEV0→ANX1	0.80 ***	0.17	ANX0→SEV1	0.79 ***	0.18
SEV1→ANX2	0.5 ***	0.14	ANX1→SEV2	0.26	0.17
SEV2→ANX3	0.73 **	0.23	ANX2→SEV3	1.03 ***	0.24
SEV3→ANX4	0.61 ***	0.11	ANX3→SEV4	0.95 ***	0.15
SEV4→ANX5	0.35	0.12	ANX4→SEV5	0.62 **	0.20
SEV5→ANX6	0.80 ***	0.15	ANX5→SEV6	0.84 ***	0.21
SEV6→ANX7	0.41 **	0.12	ANX6→SEV7	−0.16	0.11
SEV7→ANX8	0.74 ***	0.11	ANX7→SEV8	0.85 ***	0.12
SEV8→ANX9	0.56 ***	0.11	ANX8→SEV9	0.96 ***	0.19
SEV9→ANX10	0.64 ***	0.12	ANX9→SEV10	1.15 ***	0.18
SEV10→ANX11	0.63 ***	0.10	ANX10→SEV11	0.76 ***	0.18
SEV11→ANX12	0.65 ***	0.10	ANX11→SEV12	1.16 ***	0.12
**Depression**					
Cross-lagged	β	SE	Cross-lagged	β	SE
SEV0→DEP1	0.70 ***	0.14	DEP0→SEV1	0.95 ***	0.22
SEV1→DEP2	0.53	0.19	DEP1→SEV2	0.30	0.21
SEV2→DEP3	0.74 ***	0.16	DEP2→SEV3	0.98 **	0.30
SEV3→DEP4	0.40 ***	0.09	DEP3→SEV4	0.95 ***	0.19
SEV4→DEP5	0.28 *	0.11	DEP4→SEV5	0.75 *	0.30
SEV5→DEP6	0.21	0.14	DEP5→SEV6	0.66	0.34
SEV6→DEP7	0.43 ***	0.09	DEP6→SEV7	−0.16	0.21
SEV7→DEP8	0.51 ***	0.09	DEP7→SEV8	1.01 ***	0.19
SEV8→DEP9	0.34 **	0.11	DEP8→SEV9	1.19 ***	0.28
SEV9→DEP10	0.41 ***	0.12	DEP9→SEV10	1.04 ***	0.28
SEV10→DEP11	0.59 ***	0.10	DEP10→SEV11	0.89 ***	0.25
SEV11→DEP12	0.49 ***	0.07	DEP11→SEV12	1.23 ***	0.18

* *p* < 0.05, ** *p* < 0.01, *** *p* < 0.001.

**Table 3 healthcare-10-01913-t003:** Standardized parameter estimates for random-intercept cross-lagged panel models on stress (STR) and AD severity (SEV) (*n* = 19).

Cross-Lagged	β	SE	Cross-Lagged	β	SE	Covariance	β	SE
SEV1→STR2	−0.54 *	0.22	STR1→SEV2	0.28	0.26	STR1↔SEV1	0.59 ***	0.15
SEV2→STR3	−0.14	0.21	STR2→SEV3	−0.15	0.24	STR2↔SEV2	0.62 ***	0.62
SEV3→STR4	0.08	0.24	STR3→SEV4	−0.32	0.18	STR3↔SEV3	0.62 ***	0.14
SEV4→STR5	−0.35	0.23	STR4→SEV5	−0.31	0.24	STR4↔SEV4	0.61 ***	0.15
SEV5→STR6	−0.18	0.24	STR5→SEV6	−0.62 ***	0.17	STR5↔SEV5	0.56 ***	0.16
SEV6→STR7	−0.15	0.21	STR6→SEV7	0.15	0.22	STR6↔SEV6	0.26	0.21
SEV7→STR8	−0.49	0.36	STR7→SEV8	0.5	0.28	STR7↔SEV7	0.92 ***	0.04
SEV8→STR9	−0.19	0.24	STR8→SEV9	−0.22	0.27	STR8↔SEV8	0.3	0.21
SEV9→STR10	0.37	0.26	STR9→SEV10	−0.21	0.28	STR9↔SEV9	0.8 ***	0.08
SEV10→STR11	−0.35	0.37	STR10→SEV11	0.12	0.38	STR10↔SEV10	0.84 ***	0.07
SEV11→STR12	0.14	0.25	STR11→SEV12	0.29	0.26	STR11↔SEV11	0.64 ***	0.14
SEV12→STR13	0.05	0.3	STR12→SEV13	0.24	0.37	STR12↔SEV12	0.76 ***	0.09
SEV13→STR14	−0.15	0.23	STR13→SEV14	−0.17	0.23	STR13↔SEV13	0.47 **	0.18
SEV14→STR15	−0.13	0.26	STR14→SEV15	0.1	0.27	STR14↔SEV14	0.63 ***	0.14
SEV15→STR16	−0.33	0.32	STR15→SEV16	0.39	0.32	STR15↔SEV15	0.76 ***	0.09
SEV16→STR17	−0.32	0.55	STR16→SEV17	0.57	0.49	STR16↔SEV16	0.93 ***	0.03
SEV17→STR18	0.07	0.33	STR17→SEV18	−0.47	0.29	STR17↔SEV17	0.7 ***	0.12
SEV18→STR19	0.14	0.24	STR18→SEV19	−0.06	0.23	STR18↔SEV18	0.31	0.21
SEV19→STR20	0.23	0.3	STR19→SEV20	−0.26	−0.29	STR19↔SEV19	0.72 ***	0.11
SEV20→STR21	0.02	0.34	STR20→SEV21	0.03	0.33	STR20↔SEV20	0.75 ***	0.1
SEV21→STR22	0.83 **	0.29	STR21→SEV22	−0.17	0.38	STR21↔SEV21	0.83 ***	0.07
SEV22→STR23	0.25	0.26	STR22→SEV23	0.74 ***	0.22	STR22↔SEV22	0.73 ***	0.11
SEV23→STR24	−0.07	0.24	STR23→SEV24	0.19	0.25	STR23↔SEV23	0.18	0.22
SEV24→STR25	0.01	0.25	STR24→SEV25	−0.16	0.2	STR24↔SEV24	0.39	0.19
SEV25→STR26	−0.00	0.29	STR25→SEV26	−0.27	0.28	STR25↔SEV25	0.8 ***	0.08
SEV26→STR27	−0.04	0.3	STR26→SEV27	−0.13	0.28	STR26↔SEV26	0.7 ***	0.12
SEV27→STR28	−0.16	0.22	STR27→SEV28	−0.33	0.23	STR27↔SEV27	0.54 **	0.16
SEV28→STR29	0.13	0.21	STR28→SEV29	0.11	0.2	STR28↔SEV28	0.47	0.18
SEV29→STR30	0.16	0.25	STR29→SE30	−0.41 *	0.21	STR29↔SEV29	0.39	0.19

* *p* < 0.05, ** *p* < 0.01, *** *p* < 0.001.

## Data Availability

The data are not publicly available due to ethical considerations and the consents supplied by participants.

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
