# Peer review of "Prospective Analysis of the Temporal Relationship between Psychological Distress and Atopic Dermatitis in Female Adults: A Preliminary Study"

_healthcare, 2022, doi:10.3390/healthcare10101913_

Round 1

Reviewer 1 Report

This is a study aiming to assess the temporal relationship between stress, anxiety, depression and AD severity. The topic of the paper would be of interest but I am afraid that there are some flaws including: (1) the small sample size, (2) only 15 patients had a diagnosis made by a Dermatologist, (3) disease activity measures not objective (i.e. as EASI or SCORAD), (4) no information about treatments for AD/psychological intervations/ psychiatric medications that may potentially impact on POEM but also HADS and PSS-14, (5) poor quality of presentation: honestly to me the major finding of the study is not clear.

Author Response

Thank you for your comments.  In relation to point 1, we acknowledge in the paper that the sample size is small.  This study is a preliminary exploration of the associations between stress, anxiety, depression and level of AD severity in patients.  We wanted to see if this type of study was feasible with respect to ability to recruit and retain participants and if they would be able to complete measures over many time points and state this at the end of the introduction.  Although the participant number is small, we collected a huge number of data points which allowed us to report some preliminary findings.  As suggested by another reviewer, we have added ‘a preliminary study’ to the title of the paper.

In relation to point 2, 15 of the 19 had a diagnosis made by a Dermatologist as noted.  The other 4 were diagnosed by their GP.  Although this is not optimal, we do not believe that this had any impact on the results.  The POEM was completed in a similar fashion by these 4 participants as it was for the other 15 and there was no indication that these 4 participants did not indeed have AD.

In relation to point 3, the purpose of this study was to explore patients own perceptions of their disease severity, stress depression and anxiety and to use measures that patients could easily complete in their own homes during a longitudinal study.  These types of measures are necessarily subjective, and patient’s own ratings of their condition and how they felt were what we were interested in exploring.  Although more objective measures rated by clinicians are useful, it is important to understand the patient’s own view of the severity of their condition and sometimes this can be different to a clinician’s view.  The POEM is a well-validated scale which has been used across a large number of studies to investigate patients’ own perception of severity. 

It is outside of the scope of this study to also collect EASI or SCORAD data and we would not have been able to ask clinicians to record this data for their patients across all of these time points, measuring severity at the same time as their patients.  As this was a longitudinal study with data collected at a number of time points to look at cause and effect, it is also impossible to go back and collect further data. We would not be able to compare data collected now to data collected at the time (levels of stress, anxiety etc may be quite different now) and so it would necessitate a whole new study. 

We have acknowledged in the discussion section that our measure is a more subject measure and that there is evidence for its validity. We also state that clinical ratings of disease severity could be taken in future studies alongside other measures such as cortisol.

In relation to point 4, we asked participants not to take part if they were currently seeing a psychologist or psychiatrist or were on medication for a mental health issue. We omitted this from the methods section of the paper and have now included it.

In relation to point 5, we are sorry that the major finding was not clear.  We have revised the abstract and the text in the introduction and start of discussion slightly to make it clear what the aim of the study is and now state:

This study aimed to investigate whether stress, anxiety, or depression could lead to an increase in AD severity or vice versa in adults, using a longitudinal study design.

At the end of the abstract and start of the discussion section we state our main finding that there appears to be a bi-directional relationship between stress, anxiety and depression and AD severity in women. 

In the abstract we go on to say that high levels of stress should therefore be identified so that strategies can be implemented to reduce the risk of increased AD severity, and so that severity does not have an impact on psychological well-being.

We have also now included this at the end of the discussion and made clear our recommendations for the design of future studies of this nature.

Reviewer 2 Report

An original and good research investigating the relationship between stress and AD. However, number of patients is very few. I would suggest to add "a preliminary study" into the title to provoke the further studies.

Author Response

Thank you very much for this comment.  The study is indeed more of a preliminary study to also look at how feasible it is to run a longitudinal study such as this.  We have added preliminary study to the title.

Reviewer 3 Report

The authors conducted a prospective longitudinal study to evaluate the association between disease severity and psychological status in adult female atopic dermatitis (AD) patients using questionnaires. There have been many reports referring to the association of them, whereas prospective studies have been limited. The authors found that the disease severity and psychological status affect one another. Although the number of included cases is small and the results are still preliminary, the contents are of some interest. However, there are some concerns that the authors should address.

1.     The authors used POEM as the disease severity index. Surely, POEM can be used as the subjective disease severity index, whereas it can be affected by the characters of each patient. It will be better to add the objective disease severity index in the analysis. Please collect the objective disease severity index and check the association between EASI, SCORAD, or disease severity markers, such as LDH levels and TARC levels, and psychological status in some time points. Without that, this manuscript has limited impact.

2.     Table 1: “.6” in column stress and row Week 10 should be modified to “.60”.

3.     L159: “severity” should be modified to “AD severity”.

Author Response

In relation to point one, the purpose of this study was to explore patients own perceptions of their disease severity, stress depression and anxiety and to use measures that patients could easily complete in their own homes during a longitudinal study.  These types of measures are necessarily subjective, and patient’s own ratings of their condition and how they felt were what we were interested in exploring.  Although more objective measures rated by clinicians are useful, it is important to understand the patient’s own view of the severity of their condition and sometimes this can be different to a clinician’s view.  The POEM is a well-validated scale which has been used across a large number of studies to investigate severity. 

It is outside of the scope of this study to collect SCORAD data or LDH or TARC levels and we would not have been able to ask clinicians to record this data for their patients across all of these time points, measuring severity at the same time as their patients.  As this was a longitudinal study with data collected at a number of time points to look at cause and effect, it is also impossible to go back and collect further data. We would not be able to compare data collected now to data collected at the time (levels of stress, anxiety etc may be quite different now) and so it would necessitate a whole new study. 

We have acknowledged in the discussion section that our measure is a more subject measure and that there is evidence for its validity. We also state that clinical ratings of disease severity could be taken in future studies alongside other measures such as cortisol.

In relation to point two, thank you, this has been corrected.

In relation to point three, sorry I am not sure what you are referring to.  In our copy of the manuscript there is no mention of severity on line 159.  Could you please clarify by providing the full sentence.

Round 2

Author Response

There do not appear to be any further comments to address and we hope we have now addressed previous comments satisfactorily.

Reviewer 3 Report

I understand the authors' situation. Please modify the below point.

"The cross-lagged models indicated an average fit for stress and severity (χ2(1800) = 161 1677, p<0.001; CFI = 0.54, TLI = 0.67, RMSEA =0.43, 95% CI [0.42,0.46]), anxiety and AD 162 severity (χ2(1654) = 1066.98, p<0.001; CFI = 0.32, TLI = 0.21, RMSEA =0.12, 95% CI [0.09, 163 0.34]), and depression and AD severity (χ2(1435) = 1165.67, p<0.001; CFI = 0.06, TLI = 164 0.02, RMSEA =0.61, 95% CI [0.54, 0.67])."

The first "severity" should be modified to "AD severity".

Author Response

Thank you very much for clarifying this.  We have made the addition as suggested to this sentence.